# Sông Sài Gòn: Extreme Plastic Pollution Pathways in Riparian Waterways

**DOI:** 10.3390/s25030937

**Published:** 2025-02-04

**Authors:** Peter Cleveland, Ann Morrison

**Affiliations:** School of Future Environments Huri Te Ao, Faculty of Design and Creative Technologies, Auckland University of Technology, City Campus, Auckland 1010, New Zealand; peter.cleveland@autuni.ac.nz

**Keywords:** plastic waste, riparian waterways, hidden Markov model, extreme pollution

## Abstract

Plastic pollution in waterways poses a significant global challenge, largely stemming from land-based sources and subsequently transported by rivers to marine environments. With a substantial percentage of marine plastic waste originating from land-based sources, comprehending the trajectory and temporal experience of single-use plastic bottles assumes paramount importance. This project designed, developed, and released a plastic pollution tracking device, coinciding with Vietnam’s annual Plastic Awareness Month. By mapping the plastic tracker’s journey through the Saigon River, this study generated high-fidelity data for comprehensive analysis and bolstered public awareness through regular updates on the Re-Think Plastics Vietnam website. The device, equipped with technologies such as drone flight controller, open-source software, embedded computing, and cellular networking effectively captured GPS position, track, and localized conditions experienced by the plastic bottle tracker on its journey. This amalgamation of data contributes to the understanding of plastic pollution behaviors and serves as a data set for future initiatives aimed at plastic prevention in the ecologically sensitive Mekong Delta. By illuminating the transportation of single-use plastic bottles in the riparian waterways of Ho Chi Minh City and beyond, this study plays a role in collective efforts to understand plastic pollution and preserve aquatic ecosystems. By deploying a GPS-enabled plastic tracker, this study provides novel, high-resolution empirical data on plastic transport in urban tidal systems. These findings contribute to improving waste interception strategies and informing environmental policies aimed at reducing plastic accumulation in critical retention zones.

## 1. Introduction

Plastic pollution represents one of the most pressing environmental challenges of the 21st century, with devastating impacts on ecosystems, economies, and public health [1]. Global studies estimate that 70–80% of marine plastic waste originates from land-based sources, transported through rivers and waterways before entering the ocean [2,3]. These rivers act as conveyor belts, carrying enormous quantities of plastic waste from urban and industrial centers into marine environments [4]. While significant progress has been made in understanding plastic transport in oceanic systems through modeling and empirical studies [5,6,7], the transport of plastic waste through the challenging domains of urban, inland waterways remains less understood [5]. This knowledge gap limits the effectiveness of mitigation strategies aimed at intercepting plastic waste before it reaches marine ecosystems [8,9]. This study aims to bridge the knowledge gap in plastic transport dynamics by providing direct, high-resolution tracking of plastic waste in urban waterways. Unlike previous modeling or manual sampling-based approaches [10], our method allows for real-time analysis of movement patterns, retention zones, and interactions with environmental forces. By identifying plastic accumulation hotspots, this research provides actionable data for targeted waste interception and environmental conservation efforts.

The Mekong River Delta bioregion, which includes the Saigon River, exemplifies this global issue. Bioregions, defined by natural rather than political boundaries, are critical ecological units supporting diverse ecosystems and communities. The Mekong Delta, with its biologically rich wetlands, plays an essential role in Southeast Asia’s ecology [2,11]. However, this ecosystem faces immense pressure from plastic pollution, threatening biodiversity and the livelihoods of millions of people dependent on its waters [11]. The accumulation of plastic waste in these riparian ecosystems disrupts aquatic habitats [2], leading to entanglement and ingestion by wildlife, degradation of water quality, and the spread of microplastics, which further infiltrate the food chain and threaten biodiversity [10,12]. Vietnam, one of the top contributors to marine plastic pollution globally, sees between 350 g and 7.2 kg of plastic waste per capita annually entering its lakes and rivers, underscoring the urgent need for effective waste management strategies [10,13,14,15].

### 1.1. Environmental Context of the Saigon River

The Saigon River, flowing approximately 256 km through southern Vietnam, is a vital economic artery for Ho Chi Minh City, supporting trade, transport, and industry [16]. This heavily trafficked waterway hosts over 2500 large commercial vessels daily, creating a dynamic and congested environment [17]. Its role in Vietnam’s economic development is underscored by the Saigon Port, the country’s primary port, capable of hosting post-Panamax ships [16,17]. However, this economic activity comes at an environmental cost: an estimated 350,000 cubic meters of wastewater, including significant volumes of plastic waste, are discharged into the river annually [2,10,18]. This pollution, combined with intense human activity and tidal currents, renders the Saigon River an extreme environment for studying plastic transport [10,11,15,18,19].

Plastic waste in the Saigon River predominantly consists of single-use items, such as bottles and packaging materials. Studies reveal contamination levels ranging from 10 to 233 plastic pieces per cubic meter [15,20]. The complex transport dynamics of this waste, influenced by tidal flows, vessel activity, and often extreme environmental features like riparian vegetation, pose significant challenges to modeling and mitigation efforts [21]. While traditional oceanographic models have effectively predicted plastic accumulation and dispersion in marine systems, they fall short in accounting for the bidirectional flows and retention zones characteristic of tidal rivers [19,21].

### 1.2. Research Objectives and Approach

To address this knowledge gap, this study adopts a novel approach by deploying a GPS-enabled plastic bottle tracker into the Saigon River. This method builds on earlier research using drift cards [6] and oceanographic modeling [7] but introduces localized high-resolution tracking to capture the spatial and temporal dynamics of plastic transport in urban tidal systems [9]. The tracker’s deployment during Vietnam’s Plastic Awareness Month provided data across multiple tidal cycles, enabling an in-depth analysis of how tidal dynamics and external factors, such as vessel activity, influence plastic movement and retention.

This research was conducted in collaboration with Re-Think Plastics Vietnam, a non-governmental organization committed to addressing plastic pollution through innovative solutions, public outreach, and community initiatives. By aligning with Vietnam’s National Action Plan for Management of Marine Litter, which targets a 50% reduction in marine plastic waste by 2025 and 75% by 2030, this study supports national and global efforts to mitigate plastic pollution [21,22]. The primary purpose of this study is to generate high-resolution empirical data to systematically track plastic waste movement and identify key retention zones in the Saigon River, supporting targeted mitigation strategies. By employing a novel GPS-enabled tracking approach, this research provides quantitative evidence to inform localized waste management interventions and policy decisions.

## 2. Advancing Understanding of Plastic Transport in Urban Rivers

By deploying a GPS-enabled plastic bottle tracker augmented with accelerometer and gyroscopic sensors, this study offers high-resolution data transmission for remote monitoring of the spatial and temporal dynamics of plastic transport in an extreme environment [6,9]. The integration of hidden Markov models (HMMs) provides a unique probabilistic framework to analyze movement patterns and transitions between resting and transit states, capturing the complexity of plastic retention and movement under dynamic tidal conditions [21].

Focusing on the Saigon River, a challenging yet representative case of urban waterway pollution, this research highlights the critical role of tidal rivers as both transport pathways and retention zones for plastic waste [16,21]. The findings inform the development of targeted interventions, such as waste interception systems, and contribute to refining global plastic pollution models by incorporating bidirectional flow dynamics and retention behaviors [14,21]. Additionally, this study aligns with Vietnam’s National Action Plan for Management of Marine Litter, emphasizing localized, data-driven solutions to support broader sustainability goals [10,21,22,23]. By bridging gaps between global modeling and in situ observations, this research advances scientific understanding while offering practical strategies for mitigating plastic pollution [16].

The research not only contributes empirical geospatial data but also proposes innovative approaches to modeling plastic movement under complex environmental conditions. A logical thinking-frame diagram is presented below (see Figure 1). This diagram highlights the overarching problem of plastic pollution, the specific scientific gaps addressed, the methodology adopted, and the anticipated contributions to both science and policy. It serves as a roadmap to guide the reader through the progression of the study.

## 3. Materials and Methods

This project had four phases that worked in a linear procession. A collaborative interdisciplinary approach was used that had rapid cycles of hardware development, software integration, prototyping, and action research.

### 3.1. Phase One: Ideation

This phase involved scoping the most effective method for aggregating data about plastic waste in riparian waterways. After evaluating various approaches, we adopted a hybrid method that combines a drone autopilot and supporting technologies housed in a plastic bottle. This approach provided a low-cost, locally sourced solution without requiring extensive permitting or approvals, making it well-suited to the socio-technical context of the study.

### 3.2. Phase Two: Prototyping

The project underwent a phase of tandem cycles of hard and software prototyping. Iterative prototyping of hardware and feasibility prototypes of software were employed, resulting in functional prototypes that were tested and refined. This phase culminated in the development of a high-fidelity user prototype in the form of a fully functional artifact. Throughout the process, various prototyping methods, including rapid, evolutionary, and incremental approaches, were utilized to test and evaluate design decisions, leading to continuous improvements in the bottle tracker’s features and performance. The iterative nature of the prototyping process allowed for continuous feedback and adjustments until the final artifact was deemed ready for production and release “into the wild”. In parallel with the physical processes of prototyping, an initial setup of a data pipeline using industrial 4G cellular connectivity and open-source software was also focused upon to handle the tracking data efficiently.

### 3.3. Phase Three: Field Testing

A significant component of this research project was the reliability and resilience of the technology. The robustness and water tightness of the bottle tracker were tested in a controlled environment and then introduced into the environment by testing in canals, connected to the Saigon River, for set periods of time. Telemetry, data, and the digital pipeline were also tested by dry land field trips around Ho Chi Minh City to establish and map the data linkage between the bottle tracker and the cellular network it was connected to. This process also served as scouting for possible recovery sites. An action research methodology of planning, acting, observing, and reflecting was used to make any adjustments or adaptations to the bottle tracker as the field testing unfolded.

An important finding from this process was the need to maintain the correct orientation of the GPS and communication antennas for uninterrupted data transmission. The optimized arrangement relied on positive buoyancy with self-righting capability, which ensured that the tracker remained three-quarters submerged while keeping its upper components above water. After multiple prototypes, this design proved to be the most effective, allowing the tracker to float while maintaining stability even in turbulent conditions.

### 3.4. Phase Four: Plastic Bottle Tracking and Data Collection

The plastic bottle tracker was launched into the Saigon River on 1 September 2022, and the data from it were collected in real time. Daily updates and visualization on the whereabouts of the tracker were provided to the Re-Think Plastic host website. The location of the tracker was monitored to ensure that it was functional and that, if any unforeseen human interventions or incidents took place, such as interactions with large shipping, the bottle tracker could be retrieved from the waterway and repaired, if necessary. The bustling Saigon River posed significant risks of collision or entanglement for the tracker. The monitoring also ensured that the bottle tracker did not get caught up in one place. This study not only tracks plastic movement but also introduces a structured evaluation framework by analyzing key transport and retention indicators, including speed variations, resting states, and interactions with external forces such as vessel wakes and tidal reversals. These quantitative insights form the basis for defining high-retention zones and evaluating plastic transport behaviors.

### 3.5. Technologies and Development

The project employed a plastic bottle tracker to monitor plastic pollution behaviors, chosen for its prevalence as one of the most common single-use plastics found in rivers. The tracker was designed to provide daily updates during Plastics Awareness Month (see Figure 2).

The core of the system was a 3Dr Pixhawk autopilot, selected for its affordability, open-source flexibility, and autonomous functionality. This controller was equipped with sensors including a UBlox M6 GPS and Compass module, an MPU6000 accelerometer and gyroscope, and a barometer. The Pixhawk was imaged with Ardurover v4.2.3 firmware, which is part of the open-source Ardupilot ecosystem and specifically supports autonomous ground and water vehicles. Communication between the tracker and the research team was achieved using a SiK telemetry radio, later augmented with Drone Engage, a 4G-based cloud telemetry platform utilizing Vietnam’s cellular network for real-time data logging and remote internet access. To integrate Drone Engage, a Raspberry Pi Zero was used as a companion computer, connected to the Pixhawk via an industrial grade 4G dongle. This setup facilitated seamless communication with the Drone Engage server and allowed real-time telemetry data to be accessed and logged remotely. The final hardware configuration is illustrated in Figure 3, and an image of the prototype setup is shown in Figure 4.

The tracker components were housed in a watertight three-liter plastic bottle, weighted with 1.25 kg of ballast. To further protect the electronics, they were sealed inside an additional waterproof container within the plastic bottle. This design ensured the system remained operational under diverse environmental conditions. The plastic bottle’s watertightness was tested under hydrostatic pressure (9.81 kPa for three hours at a depth of one meter). A 13 dBi monopole antenna, tuned to 1800 MHz, was mounted on the top of the bottle to ensure optimal communication. Multiple iterations of the tracker were developed and tested, culminating in the release of the final prototype into the river. Photographs documenting the development process and the final prototype are presented in Figure 5a,b.

## 4. Findings

The raw telemetry data collected from the plastic tracker, including GPS coordinates, timestamps, accelerometer, and compass readings, were processed and analyzed using the MATLAB version R2024a software environment, leveraging the UAV Toolbox plugin. Data filtering and alignment were critical steps to remove irrelevant segments, synchronize timestamps at one-second intervals, and exclude noisy accelerometer and compass data.

Key metrics such as speed, distance traveled, acceleration, and turning angles were derived using mathematical models. For example, the Haversine formula was applied to calculate distance between consecutive points, while trigonometric functions were used to determine bearings and headings. These derived metrics provided an understanding of the tracker’s spatial and temporal dynamics. As this the first study we are aware of to track the movement of plastic pollution in this waterway, we did not have original data for comparative analysis testing.

To further analyze movement patterns, states of resting and transit were identified and segmented. The application of hidden Markov models (HMMs) allowed for the modeling of transitions between these states, offering insights into how tidal dynamics and external forces influenced the tracker’s behavior. Visualizations, including time-series plots and trajectory maps, were generated to explore trends, anomalies, and spatial patterns, ultimately informing the study’s findings on plastic transport behaviors in urban tidal waterways.

### 4.1. Overview of Data Collection and Analysis

Initial analysis focused on understanding the tracker’s overall behavior, including speed, gyroscopic data, and resting states.

The tracker’s movement was generally characterized by low average speeds, reflecting the influence of the tidal nature of the river [3,21]. The analysis of resting states revealed 78 periods where the tracker remained stationary, with an average resting duration of 2.40 min, highlighting how factors like tidal reversals and interactions with riverbanks can temporarily immobilize floating debris. Notably, one prolonged rest occurred at the end of the journey when the tracker settled into a tidal mud bank, demonstrating how plastic waste can become retained in such environments.

During this analysis, a notable speed spike event was identified, prompting a closer examination of this occurrence. The isolated indices of the speed spike were analyzed alongside gyroscopic and accelerometer data, providing insight into potential external forces or dynamic events, such as a ship’s wake, that could explain the sudden change in speed. This analysis suggested that the tracker experienced rotational movements, such as a barrel roll, during this period of increased velocity, as indicated by fluctuations in the gyroscopic readings.

### 4.2. Movement Dynamics (Speed, Acceleration, Distance)

The movement of the tracker was analyzed using metrics derived from GPS telemetry, including speed, acceleration, and distance traveled. These metrics provided insights into the dynamic behavior of plastic waste as it moved through the Saigon River, highlighting patterns influenced by environmental factors.

The tracker’s speed ranged from 0.0 m/s to a maximum of 2.55 m/s, with an average of 0.50 m/s. This average speed reflects the generally slow-moving nature of the river and the tracker’s dependence on tidal flows and currents for movement. Resting states, identified by a speed of 0.0 m/s, accounted for 8.79% of the tracker’s journey. The standard deviation of 0.21 m/s indicated moderate variability, with occasional spikes in speed linked to dynamic external forces, such as boat wakes or rapid currents.

Acceleration was calculated as the change in speed over time. Most of the tracker’s journey was characterized by low or negligible acceleration values, consistent with steady movement through calm water. However, periods of significant acceleration were identified during speed spikes, where external forces influenced the tracker’s behavior. These bursts of activity provided additional context for the tracker’s interaction with its environment.

The straight-line distance from the launch point to the endpoint was 26.7 km, while the total distance traveled by the tracker was approximately 205.7 km, as shown in Figure 6.

This substantial distance demonstrates the capability of the tracker to cover long stretches of the river, providing valuable insights into the transport of plastic waste. The mean distance traveled between consecutive GPS points was 4.90 m, reflecting steady, incremental movement influenced by the river’s currents. Notably, the maximum distance recorded between two points was 767.74 m, an outlier that occurred during a speed spike event, suggesting an interaction with a strong external force or rapid current.

### 4.3. Directional Behavior (Turning Angles, Bearings)

The directional behavior of the tracker was analyzed using turning angles and bearings, providing insight into how the tracker’s movement interacted with the river’s environmental conditions.

The tracker’s turning angles, calculated as the change in direction between consecutive bearings, ranged widely from 0° to 360°. The mean turning angle was 174.77°, indicating frequent sharp or near-complete directional changes, which were likely caused by currents, obstacles, or eddies in the river. This variability, with a standard deviation of 79.81°, highlights the influence of environmental forces on the tracker’s path. Periods where the tracker was stationary (resting states) were excluded from the analysis, as turning angles were undefined in these instances.

The bearings, representing the direction of movement relative to true north, revealed a predominantly south-southwest trajectory with an average heading of 187.66°. The bearing between consecutive GPS coordinates was calculated to determine the direction of movement relative to true north. This directional pattern aligns with the general flow of the Saigon River, suggesting that the tracker consistently followed the river’s natural course. Variability in bearings reflects localized deviations due to currents or interactions with obstacles.

### 4.4. Key Observations and Anomalies (Speed Spikes, Prolonged Resting)

The tracker’s journey highlighted two major anomalies: a notable speed spike event and a prolonged resting period at the end of its path. These events provide critical insights into the influence of environmental factors on plastic waste dynamics in riverine systems.

A significant speed spike was observed mid-journey, with the tracker reaching a maximum speed of 2.55 m/s, substantially higher than its average speed of 0.50 m/s. This event represented a dramatic acceleration compared to the overall movement pattern, characterized by slow and steady progression. The gyroscopic and accelerometer data provided further insights into the nature of this anomaly. Peaks in gyroscopic readings (e.g., 971, 1032, and 541 across the *x*-, *y*-, and *z*-axes) and accelerometer data (e.g., 1961 m/s^2^ along the *x*-axis) suggested that the tracker underwent intense rotational movements, such as barrel rolls, during this period which can be seen in Figure 7.

These dynamic forces were likely the result of interaction with external disturbances, such as the wake of a large ship passing through the area. Geospatial analysis of the event pinpointed its occurrence in a busy commercial zone near two large docks, close to the river’s main channel, which can be seen in Figure 8a,b.

The dense clustering of GPS points during the spike, combined with erratic pathing, indicates the turbulent nature of the environment. This suggests that anthropogenic factors, such as vessel activity, are significant contributors to the dynamic movement of plastic waste in rivers.

Across the data set, 78 distinct resting periods were identified, totaling approximately 187.27 min of stationary behavior. These events predominantly occurred in areas influenced by tidal activity or near obstacles, such as riverbanks or debris clusters. The final prolonged resting period, however, accounted for 30 h, representing an outlier event and emphasizing the significance of environmental features in plastic waste retention. This extended pause occurred when the tracker settled into a tidal mud bank near the riverbank, contrasting sharply with the average resting duration of 2.40 min observed throughout the journey. Earlier resting events were typically short and scattered, reflecting temporary immobilizations due to tidal reversals, interactions with riverbanks, or calmer water conditions. In contrast, this final resting state was likely influenced by the physical terrain and flora of the mud bank, which acted as a retention zone. These zones, particularly in mangrove areas and along tidal mud banks, become long-term plastic sinks, posing severe risks to filter-feeding organisms and altering the ecological balance of these sensitive habitats [2].

### 4.5. Hidden Markov Model (HMM) Analysis

To further understand the transitions between different movement states of the tracker, an HMM was applied. This statistical model was chosen for its ability to analyze sequential data where the system undergoes transitions between unobservable (hidden) states. By modeling stochastic processes, the HMM captures the probabilistic nature of these transitions, making it particularly suited for telemetry data where patterns in movement are influenced by complex and often indirect environmental factors, such as tides, currents, and physical obstructions. The analysis revealed two distinct states: “resting” and “transit”. The matrix, as shown in Table 1, indicated that the tracker had a 64.4% probability of remaining in the resting state once stationary and a 35.6% chance of transitioning from resting to transit. Interestingly, once in the transit state, the probability of remaining in that state was 100%, suggesting that external forces were required to initiate movement but that continuous flow, likely driven by currents and tides, sustained the transit state.

The HMM emission matrix provided additional insights into the tracker’s behavior. While in the resting state, the tracker predominantly exhibited medium speeds, which likely reflected subtle oscillations caused by localized turbulence. In contrast, during the transit state, medium and high speeds were more commonly observed, corresponding to steady or accelerated motion driven by dynamic external forces. This process is visualized in Figure 9.

These results provided valuable insights into the dynamics of plastic transport, especially highlighting how transitions from resting to transit are less likely without significant external influences like tidal shifts or boat wakes. The consistent retention in the resting state also aligns with observed resting periods, suggesting that floating debris can become trapped in particular zones before external forces shift them back into motion. The prolonged retention of plastic waste in certain areas indicates that these locations may act as chronic pollution hotspots, leading to bioaccumulation of plastics in local fauna and persistent disruptions to aquatic food webs [3,5,10,11].

## 5. Discussion

In this study, we designed, fabricated, and deployed a GPS-enabled tracker to understand the movement of a single-use plastic bottle, extending prior research [9,16,21], within the Saigon River’s urban tidal environment. The tracker was developed to capture detailed movement data, including GPS location, speed, and orientation, augmented by gyroscopic, compass, and accelerometer readings. Field deployment involved releasing the tracker into the river system during Vietnam’s Plastic Awareness Month, gathering data across multiple tidal cycles to capture the complex transport dynamics of plastic debris, which provides critical scientific evidence on how plastic waste is transported through urban waterways.

The purpose of this study was to systematically track plastic waste movement and identify key retention zones in the Saigon River to support targeted mitigation strategies and generate high-resolution empirical data that reveal how plastic waste is transported, retained, and influenced by both natural and anthropogenic forces. This study highlights the broader problems [1,2,10] with the escalation of plastic pollution growth. Even now we have environmental problems in managing the current level of output into the oceans [3,4,11] and accumulation on shorelines [15,19]. This research builds on data [9,18,20] and contributes to marine debris modeling more generally [8], adding information [19,20] and building on prior work identifying threats to the river [2,13,21]. These findings indicate that plastic accumulation in high-retention areas is not random but governed by tidal reversals, vessel activity, and structural barriers. By analyzing the collected data, this study provided critical insights into the transportation behaviors of plastic waste, highlighting the significant impact of tidal reversals, external disturbances, and localized environmental conditions on plastic movement and retention. As the first study of its kind in this region, we acknowledge that no comparative data sets exist; however, we have established a systematic tracking approach to address this gap. By illuminating the transportation of single-use plastic bottles in the riparian waterways of Ho Chi Minh City and beyond, this study achieves its purpose in highlighting where accumulation points are, which plays an important role in determining where best to begin to collect pollution and in collective efforts to understand plastic pollution and preserve aquatic ecosystems. The identification of these retention zones is crucial for eradicating plastic waste from this vital waterway and informing effective waste removal efforts.

### 5.1. Findings and Interpretations

The tracker’s journey provided insights into the behavior of plastic waste in tidal environments, characterized by slow movement and frequent resting states. The tracker’s low average speed of 0.50 m/s and 78 distinct resting periods, with an average duration of 2.40 min, underscore the significant role of tidal reversals and interactions with riverbanks in immobilizing plastic waste [2,3,21]. The final resting period, spanning 30 h in a tidal mud bank, highlights how specific environmental features can act as long-term retention zones, preventing plastic from reaching open waters [16].

These observations align with prior studies emphasizing the retention potential of tidal systems, adding empirical evidence to support their role as semi-permanent repositories for plastic debris [2,3,16,21]. The prolonged resting period highlights the capacity of natural features, such as mud banks, to trap floating debris. This retention has important implications for understanding plastic waste accumulation in riparian systems and for designing targeted waste removal interventions. The persistence of the tracker in this state underscores the need to monitor such zones as key sites for intervention strategies, including the installation of plastic capture systems.

A notable anomaly in the data was the speed spike event, where the tracker experienced a sudden acceleration to 2.55 m/s, far exceeding its average velocity of 0.50 m/s. This event, supported by gyroscopic and accelerometer readings, revealed intense rotational dynamics and sudden shifts in speed and orientation, indicating a momentarily chaotic environment influenced by external forces. Analysis of the gyroscopic data showed rapid movements consistent with what would occur during interactions with large-scale vessel wakes or high-turbulence zones. This dynamic behavior aligns with the expected effects of passing ships in busy commercial waterways, where wake turbulence can displace floating objects with significant force. This dynamic behavior aligns with the expected effects of passing ships in heavily trafficked commercial waterways, where wake disturbance can displace floating objects with significant force. The Saigon River, as a vital economic artery for Ho Chi Minh City [17,24], underscores this contrast of scale and activity. Navigated daily by over 2500 large commercial vessels, including post-Panamax ships, the waterway is highly dynamic and congested [16].

These massive vessels, essential for Vietnam’s economic development, produce significant wake turbulence that interacts with the comparatively minute scale of a single-use plastic bottle tracker. This stark contrast between the scale of shipping infrastructure and the vulnerability of floating plastic debris highlights the disproportionate impact of human activities on the movement of waste in urban waterways. The bustling traffic amplifies the complexity of plastic transport as human-engineered forces interact with natural tidal dynamics to create unpredictable transport patterns, further emphasizing the Saigon River’s role as both a transport pathway and retention zone for plastic debris.

The geospatial context of the speed spike event further supports this reading, as the anomaly occurred near two heavily trafficked commercial docks situated along the Saigon River’s main channel. This zone, characterized by constant vessel movement and industrial activity, highlights the significant role human interventions play in disrupting the natural transport patterns of plastic waste. Large vessels not only generate wake turbulence but also create secondary effects, such as rapid water displacement, increased flow velocities, and pressure changes, all of which can contribute to sudden, erratic movements of floating debris [14]. This underscores the unpredictable nature of plastic transport in regions where human and natural dynamics intersect. Unlike in more predictable open-water environments, urban waterways like the Saigon River present a complex interplay of anthropogenic and hydrodynamic forces [21]. For example, vessel-induced turbulence may temporarily dislodge plastic debris from retention zones, altering its trajectory and influencing its eventual accumulation site. This unpredictability complicates the task of modeling plastic transport and highlights the need to incorporate localized human impacts into pollution models.

The speed spike event also raises important questions about the broader implications of anthropogenic activities for plastic pollution pathways [15]. In heavily trafficked urban waterways, vessel wakes may serve as both a dispersal mechanism and a temporary accelerant for plastic debris, redistributing waste in unpredictable ways. This suggests that mitigation strategies should account for these interactions, particularly in zones with high vessel density. Targeted monitoring of such areas could provide critical insights into the role of human activity in shaping plastic pollution dynamics and inform the placement of waste interception systems to counteract these effects. Our findings indicate that plastic accumulation in high-retention areas is not random but governed by predictable hydrodynamic and anthropogenic factors. By systematically identifying these retention zones, this study provides a foundation for targeted plastic removal interventions, aligning with broader efforts to mitigate plastic pollution in urban waterways.

The application of a hidden Markov model (HMM) provided a probabilistic framework for analyzing the tracker’s movement states. The model identified two distinct states: “resting” and “transit”. The tracker exhibited a 64.4% probability of remaining in the resting state and a 35.6% chance of transitioning to transit, indicating that external forces, such as tidal shifts or boat wakes, were often required to initiate movement. Once in transit, the tracker consistently remained in motion, driven by currents and sustained flow [9,21]. The HMM findings highlighted the intricate balance between stationary and dynamic phases in plastic transport, offering an understanding of how tidal environments govern plastic retention and movement. This is the first use of hidden Markov models (HMMs) for this purpose, and we believe it has added to the available data collection and analysis methods for the scientific research.

### 5.2. Implications for Pollution Models and Broader Context

The findings from this study challenge traditional models of plastic transport, which often assume unidirectional flow and fail to account for the complexities of tidal systems. The results demonstrate that bidirectional flows, frequent resting states, and retention zones, such as tidal mud banks, significantly influence the movement and accumulation of plastic waste [14,16,21]. These dynamics delay plastic transport to open waters, indicating that tidal rivers and estuaries may act as semi-permanent repositories for plastic debris [2,15].

Integrating these findings into global plastic pollution models could improve their accuracy, particularly for tidal rivers and estuaries. Models incorporating bidirectional flows and retention dynamics would better predict plastic accumulation hotspots and the timing of plastic emissions into oceans. This study underscores the importance of identifying and monitoring retention zones, which could inform the design of targeted cleanup efforts and capture systems [20,21]. By highlighting the interplay between natural and anthropogenic factors, this research contributes to a more comprehensive understanding of plastic pollution pathways in riparian waterways and their implications for marine debris management.

### 5.3. Limitations and Scope of Findings

While this study offers a detailed case study of plastic transport in the Saigon River, it is context-specific, and the findings may not be directly applicable to all tidal rivers without considering their unique hydrodynamic and environmental conditions [21]. Additionally, the tracker design itself posed limitations, such as requiring a specific buoyancy configuration to keep the GPS antenna pointed skyward, which may influence its behavior differently from unmodified plastic waste. The results presented here are best understood as illustrative of potential trends rather than universally applicable findings, though the principles observed can inform broader research in similar tidal environments.

### 5.4. Contributions and Future Work

This research makes contributions to the growing body of knowledge on plastic transport dynamics in riparian and tidal environments. By employing a low-cost, field-based approach, this study demonstrated the efficacy of using a GPS-enabled plastic tracker augmented with accelerometer data and hidden Markov model (HMM) analysis to achieve a granular understanding of plastic retention and movement under complex tidal conditions [11,21]. The integration of these methods provides a novel framework for analyzing plastic waste behavior, advancing methodologies for studying pollution pathways in dynamic environments.

Furthermore, this research emphasizes the value of in situ observations in real-world, heavily polluted challenging environments like the Saigon River. These findings highlight the limitations of traditional plastic transport models that rely on unidirectional flow assumptions, underscoring the need for models that account for bidirectional flows, tidal reversals, and retention zones. By addressing these gaps, this study contributes to the refinement of plastic pollution models and offers practical insights for targeted cleanup efforts and policymaking [23,25].

Looking ahead, future research could extend the scope of this study by deploying multiple GPS-enabled trackers across a range of tidal and non-tidal river systems. This expanded approach could allow for a comparative analysis of transport dynamics under varying environmental conditions, including areas of high plastic accumulation, such as collection and diversion points along the Saigon River. These studies could inform the design of localized interventions and enhance global understanding of plastic retention and movement [26], identifying more opportunities for efficient redirection and removal of waterway waste.

Another promising avenue for future work is the examination of how different types of plastic and waste, varying in buoyancy, rigidity, and shape, interact with tidal flows and external forces. By deploying trackers with varying buoyancy characteristics, researchers could better understand the specific behaviors of diverse plastic types under natural conditions. This knowledge could guide waste management strategies and inform efforts to design more effective pollution control measures in similar tidal waterways [10]. Future studies could also focus on identifying optimal locations for installing plastic waste capture systems by analyzing the movement and retention patterns of different plastic types in areas prone to accumulation.

As part of the broader findings, this study emphasizes the importance of fostering informed awareness to bridge the gap between scientific research and community action on plastic pollution. In addition to collaborating with Re-Think Plastics Vietnam and supporting their initiative during Plastic Awareness Month, the research findings will be disseminated in formats suitable for general public reading in both English and Vietnamese. This approach ensures that local researchers, students, and non-governmental agencies have access to data sets that directly map and inform how discarded single-use plastic bottle waste moves through the inland waterway of the Saigon River. By doing so, we aim to foster community engagement, support localized environmental action, and strengthen awareness around plastic pollution’s impact on Vietnam’s critical ecosystems.

## 6. Conclusions

This study investigated the movement and retention dynamics of a single-use plastic tracker in the tidal waters of the Saigon River, shedding light on the broader challenge of plastic pollution in riparian ecosystems. Using a combination of low-cost GPS-enabled tracking, accelerometer, and gyroscopic sensors and hidden Markov model (HMM) analysis, we provided a detailed understanding of the tracker’s behavior, including speed, resting states, and responses to external influences. The results revealed the critical role of tidal dynamics in governing the slow average speeds and frequent resting periods of floating debris. A notable speed spike event, attributed to vessel-induced turbulence, demonstrated the significant impact of anthropogenic activities on plastic transport. The HMM analysis further provided insights into transitions between resting and transit states, emphasizing the potential for retention zones in tidal waterways to act as semi-permanent repositories supporting efficient collection of plastic waste.

By illuminating the transportation of single-use plastic bottles through the riparian waterways of Ho Chi Minh City, this study contributes to collective efforts to understand and mitigate plastic pollution, supporting the preservation of aquatic ecosystems. The findings align with prior research on plastic retention in estuaries, underscoring the intricate interplay between tidal cycles, river discharge, and human activity in influencing plastic transport. This study advances scientific understanding by providing a structured approach to tracking and evaluating plastic waste transport in tidal river systems. By integrating GPS tracking with empirical data analysis, it establishes a reproducible methodology for assessing plastic retention patterns, informing cleanup strategies, and refining plastic pollution models.

The contributions of this work are threefold: it demonstrates the practical application of GPS-enabled tracking for monitoring plastic waste, highlights the importance of integrating tidal dynamics into plastic transport models, and offers an innovative approach to analyzing movement patterns using HMMs. Future research could build upon this study by deploying similar trackers across diverse river systems, both tidal and non-tidal, to explore variability in transport dynamics. Investigating how different plastic types, with varying buoyancy and rigidity, interact with environmental forces could provide further insights into plastic behavior. Additionally, identifying and prioritizing retention zones and collection points along waterways like the Saigon River could enhance the efficiency of cleanup efforts and contribute to long-term pollution mitigation strategies.

## Figures and Tables

**Figure 1 sensors-25-00937-f001:**
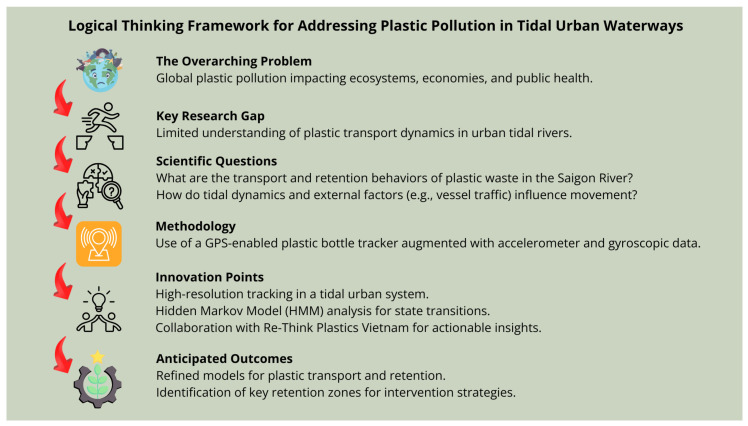
Logical thinking-frame diagram. Arrows indicate progression.

**Figure 2 sensors-25-00937-f002:**
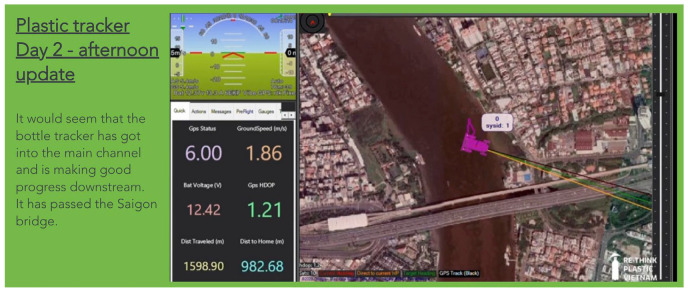
Day 2 daily update (screenshot).

**Figure 3 sensors-25-00937-f003:**
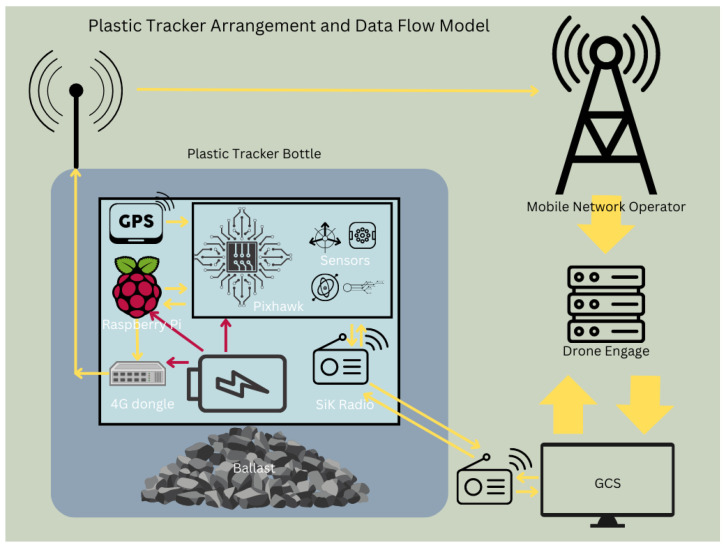
Technology arrangement and data flow (diagram). Arrows indicate data flow direction.

**Figure 4 sensors-25-00937-f004:**
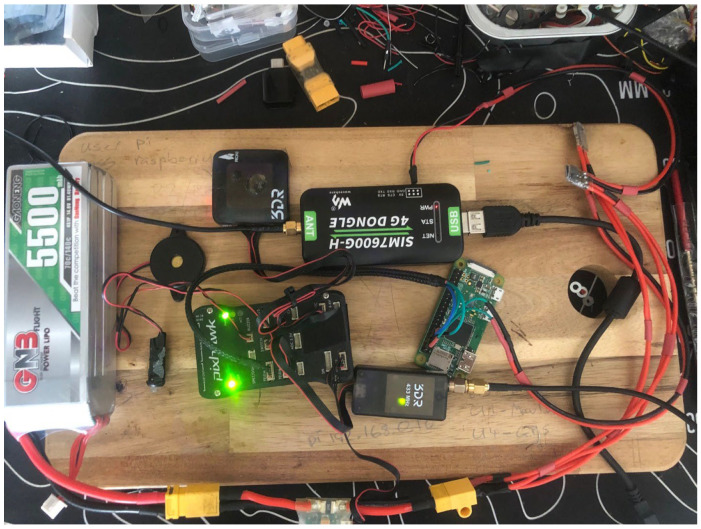
Final technology arrangement before installation in bottle (photo).

**Figure 5 sensors-25-00937-f005:**
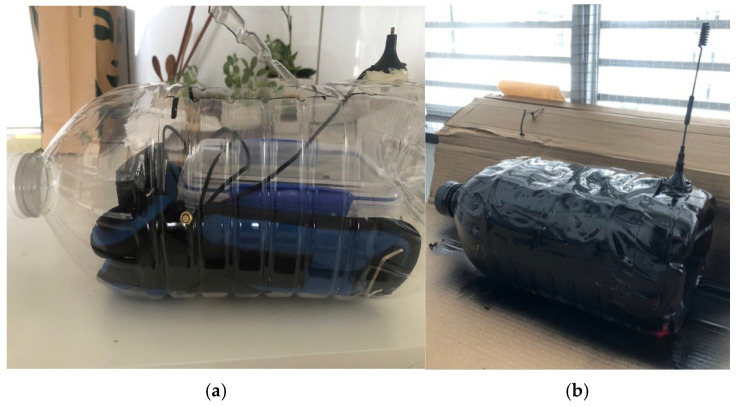
(**a**): Plastic tracker development with final ballast and sensor, housing arrangement. (**b**): Plastic Tracker final prototype as released into the river (photos).

**Figure 6 sensors-25-00937-f006:**
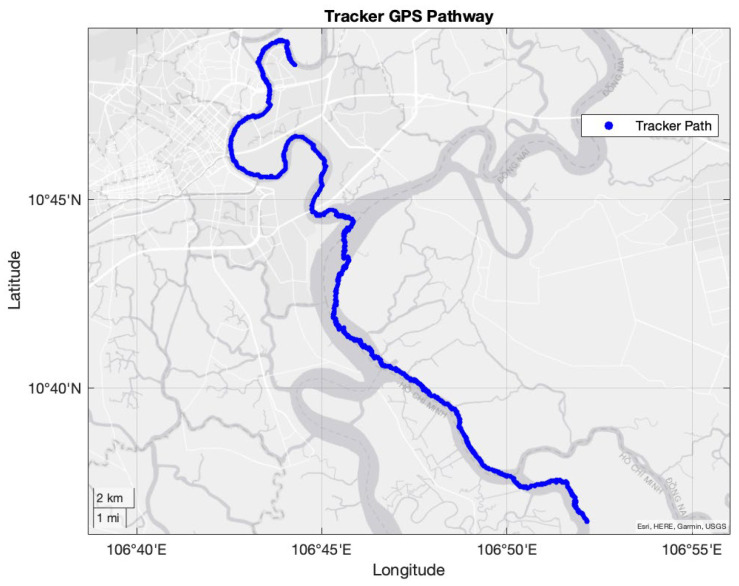
Pathway of tracker from launch point until end point (image).

**Figure 7 sensors-25-00937-f007:**
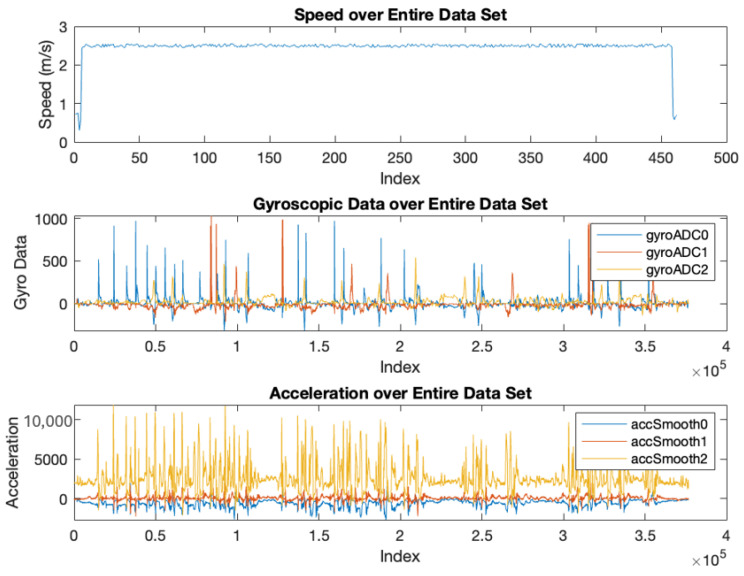
Speed, Gyroscopic, and Accelerometer Data during Speed Spike Event (image).

**Figure 8 sensors-25-00937-f008:**
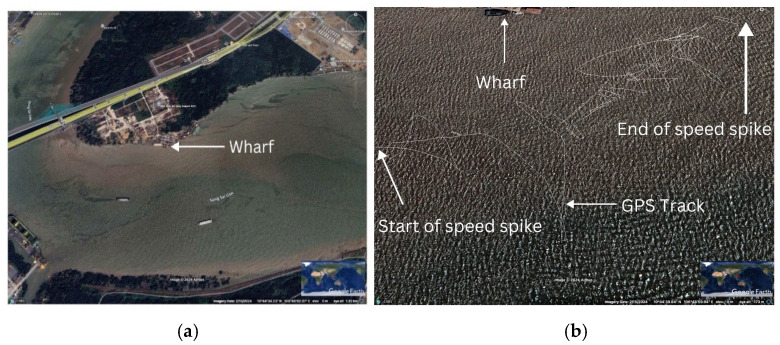
(**a**): Overview of Speed Spike Event Location. (**b**): Detailed View of GPS Track During Speed Spike (images).

**Figure 9 sensors-25-00937-f009:**
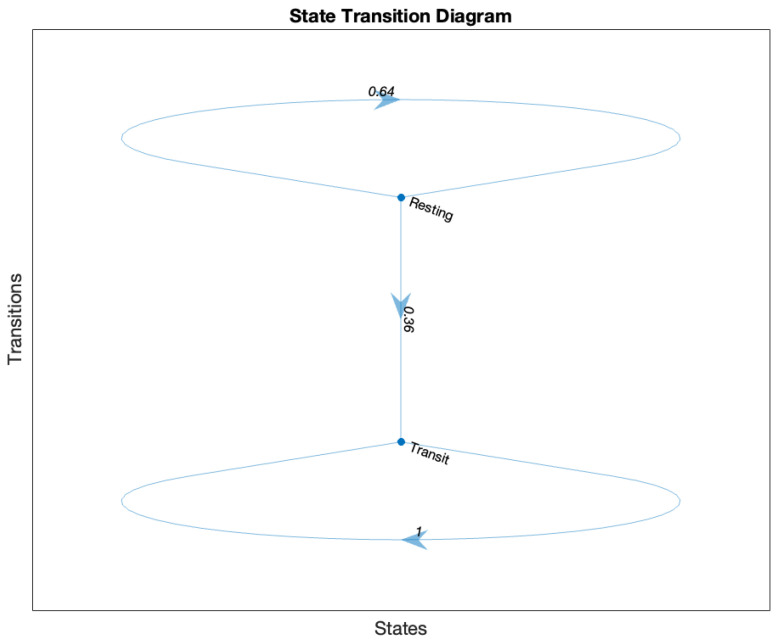
State Transition Process. Arrows indicate progression.

**Table 1 sensors-25-00937-t001:** Estimated Transition and Emission Matrices (table).

Matrix	State Transition/Emission	Probability
Estimated Transition Matrix	Resting → Resting (State 1 to State 1)	0.644
	Resting → Transit (State 1 to State 2)	0.356
	Transit → Transit (State 2 to State 2)	1
Estimated Emission Matrix	Low Speed while Resting	0.1275
	Medium Speed while Resting	0.7934
	High Speed while Resting	0.0792
	Low Speed while in Transit	0.0165
	Medium Speed while in Transit	0.6152
	High Speed while in Transit	0.3684

## Data Availability

The data presented in this study are available on request from the first author. The data are not publicly available due to the costs and resources involved in making the data set legible enough to be publicly accessible. We will make the data publicly available in the next iteration of this project.

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
