# Peer review of "Sông Sài Gòn: Extreme Plastic Pollution Pathways in Riparian Waterways"

_sensors, 2025, doi:10.3390/s25030937_

Round 1

Reviewer 1 Report

Comments and Suggestions for Authors

1. The paper has made a sufficient literature review on river pollution, without highlighting the key points. It is suggested to summarize the main innovation points and main research contents of this paper.
2. Plastic bottles are used as tracking devices, which are relatively novel and environmentally friendly. It is suggested that a logical thinking-frame diagram be sorted out to summarize scientific problems.
3. See that the author has done a lot of work, but the content is not logical enough. It looks more like a project report than an academic paper.
4. The paper needs to be reformatted and polished. Some formulas are large, some are small, and there are typos and incorrect use of symbols.
5. The table is overflowing, and it is recommended to readjust.
6. The paper aims at applying innovation, but the innovation point is not prominent enough, and the method application is not clear.

Comments on the Quality of English Language

The English could be improved to express the research more clearly.

Reviewer 2 Report

Comments and Suggestions for Authors

The manuscript «Song Sai Gon : Extreme Plastic Pollution Pathways in Riparian Waterways » submitted by Cleveland and Morrison reports the conception, development, and released of a plastic pollution tracking device. The results show the plastic tracker’s journey in the Saigon River. The results show the behaviours of plastic pollution and help to better preserve the aquatic ecosystems. This is, to my knowledge, the first paper showing the journey of a plastic bottle throw the Saigon River to the delta, it is an impressive work. However, the paper needs deep rearrangements prior publication. I recommend a Major Review

The first point concerns the organization of the paper. The introduction gives good global information on the Saigon River context. However, between the Introduction (section 1) and the section 2 (environmental context) we can observe some repetitions on several data. I give some examples below but I recommend the authors to deeply reorganize the introduction in order to give global information, then give the context of Saigon River and then introduce the problematic and how this paper will permit to resolve the scientific question. In addtion, at the end of the introduction, you do not give the objectives of the paper, it is hard to follow. I suggest a fusion of part one and two and deeply reorganize the introduction and give at the end clear objectives and how the authors can solve the question.

The second point is the section 3. I recommend to entitle it material and methods. In addition, you present results that failed (sections 3.1 – 3.4), why ? I think it is not necessary to present it. Just give the facts that work in your study. Or, if you really want to present this, explain clearly why ? Some other authors tried this before ?

Concerning the other sections, 4 to 11, I think it is the result section. However, it is too dense. You need to reorganize the results. You show that you investigates several parameters such as speed, angle, gyroscopic data. You need to simplfy all of this to give the essential results. Why did you use the Hidden Markov Model ? The section 11, findings is the abstract of all the other section. I think, you should begin with this section to simplify all the other section from 4 to 10. Do tables, resume graphs. 

The discussion part is interesting and permits to better understand the results and to compare them with previous studies. However, compared to the result section, it is not equilibrated. In addition, why you divide the discussion section in 12.1 to 12.7 ? I think you can just do three or four parts in this section like the results, the limitations and future directions. I really think that you need to rewrite the result section in order to understand easily all the study that is to my knowledge unique and very interesting. 

Below are presented my recommendations on focused areas of the manuscript. 

Line 26 : Begin an introduction with the aim of the paper is, to my point of view, not a good firt introduction sentence. You need to give very global vision first.

Line 53 : you give information on the number of vessels but you also give this info in the part 2, you have repetition, need reorganization. 

Reviewer 3 Report

Comments and Suggestions for Authors

Given that the purpose of the study is to provide useful information for cleaning the riparian waterways from plastic pollution, it is recommended to conduct future studies with different types of plastic waste to identify potential locations along the river path that are most suitable for installing plastic waste capture systems.

Round 2

Reviewer 1 Report

Comments and Suggestions for Authors

By tracking the development status of plastic bottles through positioning, describe the impact on the biological environment and ecological environment, the purpose and function are not clear.

The purpose and evaluation indicators are not clear. This means that the scientific issues have not been extracted out. Only the applications and key technologies have been discussed.

Comments on the Quality of English Language

There is still room for improvement.

Author Response

Please see the attachment in response to the reviewer's recommendations

Reviewer 2 Report

Comments and Suggestions for Authors

The manuscript «Song Sai Gon : Extreme Plastic Pollution Pathways in Riparian Waterways » submitted by Cleveland and Morrison reports the conception, development, and released of a plastic pollution tracking device. The results show the plastic tracker’s journey in the Saigon River. The results show the behaviours of plastic pollution and help to better preserve the aquatic ecosystems. This is, to my knowledge, the first paper showing the journey of a plastic bottle throw the Saigon River to the delta, it is an impressive work. The authors answered to all my inquiries and this manuscript is now ready for publication. 

The first point concerning the organization of the paper has been improved. The introduction gives good global information on the Saigon River context and has been reorganized with new figures as well. Clear objectives has been added.

The material and methods section has been implemented clearly and the result section as well. The discussion part is interesting and clearly added to the paper and clearly equilibrated with the rest of the sections.  

Thank you to have clearly reorganized the paper, it is ready for publication.

Author Response

Thank you so much for your helpful review. We really appreciate your advice and recommendations and are genuinely pleased with the improvements to the paper thanks to your generous insights and input.